# Students’ Perceptions in Online Physical Education Learning: Comparison Study of Autonomy, Competence, and Relatedness in Saudi Students during the COVID-19 Lockdown

**DOI:** 10.3390/ijerph192215288

**Published:** 2022-11-19

**Authors:** Mohamed Frikha, Nesrine Chaâri, Nourhen Mezghanni, Majed M. Alhumaid, Mohammed S. Alibrahim

**Affiliations:** 1Department of Physical Education, College of Education, King Faisal University, Al-Ahsa 31982, Saudi Arabia; 2Research Laboratory-Education, Motricity, Sport and Health (LR19JS01), High Institute of Sport and Physical Education, Sfax University, Sfax 3000, Tunisia; 3Department of Sport Sciences, College of Education, Taif University, Taif 21944, Saudi Arabia

**Keywords:** online learning, physical education, self-determination theory, psychological need satisfaction

## Abstract

The interest in the efficiency of online learning was and remains a major concern to researchers especially during the worldwide pandemic crisis (COVID-19). Nonetheless, there is a lack of studies focusing on students’ perceptions in online physical education (PE) learning sessions. Based on the self-determination theory (SDT), the present investigation aimed to explore psychological need satisfaction (PNS) to understand the autonomy, competence, and relatedness of Saudi PE students during the online sessions undertaken during the COVID-19 pandemic lockdown. PE students (N = 321, 161 females and 160 males) completed an online questionnaire composed of demographic characteristics, grade point average (GPA), sleep and physical activity (PA) habits, and the physical education autonomy relatedness competence scales (PE-ARCS). A t-test and one-way ANOVA were conducted and revealed that female students presented higher PNS compared with males. Students practicing PA had higher autonomy values than those not practicing PA (*p* = 0.001). However, no differences were recorded concerning competence and relatedness perceptions. The groups practicing walking, aerobic exercise, muscular training, and specialty training had higher values in autonomy and relatedness. The groups that slept for more than six hours a night, had previous experience with online learning, and had a GPA of more than three recorded higher PNS values. Correlation analysis showed high interdependence of the three PNS variables as well as with the variables of gender, experience with online learning, sleep hours, and type of PA practiced, but not with GPA or frequency of PA practice. The PNS values regarding online PE sessions were (i) higher in female students compared with males and (ii) related to previous experience in online learning, GPA, sleep habits, and type of PA. Walking, aerobic exercise, muscular training, and training in a specialty affected both autonomy and competence perception; however, relatedness was mainly affected by walking activity. Therefore, it is necessary to support ICT knowledge of students with low GPAs and to encourage them to adopt balanced sleep and physical activity habits to increase their perceptions of autonomy, competence, and relatedness in online PE lessons.

## 1. Introduction

Online learning courses have been widely adopted by higher education institutions as another method to substitute traditional classroom instruction, mainly during the COVID-19 pandemic when instructional institutions (e.g., colleges and universities) were unable to pursue education in a traditional in-class setting [1]. In Saudi Arabia, online education for all schools and universities was imposed on 8 March 2020 to ensure the safety of students and teachers [2]. Thus, teaching in universities was applied via different online platforms, such as Blackboard, Microsoft Teams, and Zoom. This abrupt transition of instructional mode was indeed not without difficulties for administrations, teachers, and students [3].

In general, online learning is defined as any setting that uses information communication technology (ICT) to improve the quality of the teaching–learning process [4,5]. It was demonstrated that online learning is one of the most used forms of learning in the higher education context [6,7]. It provides different experiences to the traditional classroom, as communication is via the Internet and the social dynamics of the learning environment are different [8]. According to Alkinani [9], online education is characterized by the separation of teachers and learners (which distinguishes it from face-to-face education), the use of a computer network to present or distribute educational content, and the provision of two-way communication via a computer network so that students may benefit from communication with one another, teachers, and staff.

However, considerable concerns and problems have ensued, particularly regarding the quality of online education, the acceptance of this instructional modality by learners, and the specificity of the learning subject. The use of online learning in physical education (PE) creates a range of challenges too, which are essentially related to the lack of interaction. While some general objectives may be reachable, such as physical or motor development, others, such as social or affective objectives, seem to be difficult to attain. Theories suggest that learning is promoted or enhanced (1) when students are actively involved in the learning process, (2) when assignments reflect real-life contexts and experiences, (3) when critical thinking or deep learning is promoted through applied and reflective activity [10], and (4) when the learning situation enhances social interaction [11]. In general, the purpose of the learning situation is to provide students with the opportunity to interact socially with others. They may interact not only with teachers but with peers as well. Learning was recognized as a social activity long before the onset of online education [12], and the conceptualization of learning as a social and interactive activity provides a rationale for promoting a sense of learning community. Learning communities support collaborative learning, which may enhance the active exchange of ideas within small groups and increase the interest of participants [13]. The argument that a sense of community promotes learning outcomes and student satisfaction with learning experiences is common in the literature [12,14]. Therefore, the sense of belonging of students (including to the school), that they matter to one another and to the group, that they have duties and obligations to the school and one another, and that they have a shared conviction that the educational needs of members will be met through their commitment to common goals [15]. Thus, the learning community (or classroom) seems to be consistent with socio-constructivist educational practices, which emphasize learning as a social and interactive activity [11].

Focusing on online learning efficiency requires investigations of student perceptions of this modality of instruction. In the last decade, much research has been conducted on motivation and psychological needs satisfaction (PNS) in PE students of different academic levels: primary school [16,17] and middle school [18,19]. However, and to the authors’ best knowledge, no study has investigated PNS among university PE students in an online learning context. Therefore, the present study examined whether accessing students’ learning perceptions in online PE can in itself foster their PNS. This investigation aimed to assess the perception of students regarding PE and to compare their autonomy, competence, and relatedness during the undertaken online lessons according to the self-determination theory (SDT) [20]. This may be of great interest for teachers to better motivate students in online PE lessons as well as in their didactic and/or pedagogical choices when structuring the teaching–learning process. The term SDT refers to ‘‘a quality of human functioning that involves the experience of choice. [It is] the capacity to choose and have those choices be the determinants of one’s actions” [21]. SDT proffers that humans have three universal and basic needs: autonomy (a feeling of personal agency and ability to make your own decisions in different activities), competence (a feeling of effectiveness when interacting with the environment and engaging in optimally challenging tasks), and relatedness (the feelings of social inclusion and closeness) [21,22]. It was demonstrated that fulfillment of these needs guides and provides energy when engaging in certain physical activity (PA) behavior [20]. Previous studies showed that students’ PNS have a positive impact on enjoyment in the PE context. Indeed, enjoyment was linked to the perception of autonomy [23], competence [24], and social relatedness [25].

## 2. Materials and Methods

### 2.1. Sample Size

There are 29 government universities in Saudi Arabia. Of these, seven contain PE, sports science departments, and one college. Three of these seven institutions were geographically randomly chosen so that the western, central, and eastern geographical regions in Saudi Arabia were accounted for. A total population in the three universities was set at 705 students (382 males and 323 females). Assuming a prevalence (p) of 50% with a required confidence level of 95%, a margin of error (e) of 5%, and an associated z-score of 1.96, the sample size (n) was calculated according to the following formula [26]:n = [z^2^ × p × (1 − p)/e^2^]/[1 + (z2 × p × (1 − p)**/**(e^2^ × N)]
and was set at 249 participants. Considering a dropout rate of 20% [27], a minimum of 299 students were requested to participate. Nonetheless, and to ensure better student participation, the online questionnaire link and the informed consent were sent to almost all regular PE students in the selected universities (705 requests) by emails after contacting the Information Technology Deanship in each University. In total 321 positive responses (160 males and 161 female) were received and were considered as the final sample size of the study. 

### 2.2. Participants

PE students belonging to three universities (King Faisal University, KFU; King Saud University, KSU; and Taif University, TU) participated in the study from 13 October to 18 November 2020. All of the students experienced online learning sessions for at least three months and practiced term and final evaluations via the Blackboard platform. The inclusion criteria were (i) study regularity, (ii) the presence of practical and theoretical courses in the student’s timetable, and (iii) answering all questionnaire questions. All respondents were fully informed about the objective of the study and voluntarily agreed to participate.

### 2.3. Questionnaire

Based on the PE autonomy relatedness competence scales (PE-ARCS) of Sulz et al. [28], we developed an Arabic version of the questionnaire, which consisted of three parts and took approximately 15 min to complete. The first part collected demographic data (age, gender, weight, height, regularity of study in university, grade point average (GPA; 3 < GPA ≤ 5 and 1 < GPA ≤ 3), and experience in online learning. In the second part, the participants were asked to provide information about their sleeping routine (daily sleeping hours: < 5 h; between 6 and 9 h, and >10 h), PA habits (activities undertaken not less than 30 min each time: walking, aerobics, muscular reinforcement, specialty training, other activities), and frequency of PA practice per week (one time, two times, three times, four times and more) during the COVID-19 lockdown period. The third part consisted of 12 items for which the students had to rate their perceptions during online classes regarding autonomy (items 3, 6, 9, and 12), competence (items 2, 5, 8, and 11), and relatedness (items 1, 4, 7, and 10). A Likert scale was used (1 = strongly disagree, 2 = disagree, 3 = neutral, 4 = agree, 5 = strongly agree).

To ensure that the participants understood the questions, the English version of the PE-ARCS [28] was translated and adapted to the Saudi context following the recommendations of Beaton et al. [29]. Two bilingual translators first translated the questionnaire into Arabic and a standardized version was extracted. Two English speakers of Arab origin then back-translated this version into English, and three bilingual experts reviewed both versions (Arabic and English). The necessary adjustments requested by the experts were made. Finally, the reliability of the Arabic version of the PE-ARCS was examined separately and as part of the overall questionnaire. All Cronbach’s alpha values were satisfactory, with 0.727 for autonomy, 0.823 for competence, 0.616 for relatedness, and 0.902 for the all-item questionnaire.

### 2.4. Data Analysis

The independent variables were gender, experience with online learning, university, grade point average (GPA), hours of sleep, physical activity practice, and type of physical activity (PA). Statistical analysis was performed using SPSS V.26 (IBM, Armonk, NY, USA). Descriptive data were summarized as means and standard deviations or proportions of the total population. Normality was tested using histograms and absolute values of skewness, and all values were < 2 [30]. Student’s t-tests and one-way ANOVAs were used to test for significant differences between the groups, stratified by gender, university, experience with online learning, hours of sleep, and PA for all continuous variables. The Bonferroni post hoc test was used to test the between-group differences. Pearson’s correlation was performed to determine the relationships between autonomy, competence, and relatedness, as well as with the studied variables. The significance level was set at *p* < 0.05.

## 3. Results

### 3.1. Sociodemographic Characteristics

In total 321 PE students (160 male and 161 female) participated in the study with a response rate of 45.5%. Table 1 shows the descriptive statistics of the variables for gender, age, body mass index, universities, previous experiences to online learning and GPA.

### 3.2. Effect of the Variables of Gender and Physical Practice

The results related to the effects of the variables of gender and physical practice on autonomy, competence, and relatedness are presented in Table 2. 

Concerning the gender variable, the statistical analysis showed higher values in female compared with male students on autonomy, competence, and relatedness (*p* = 0.037, *p* = 0.001, and *p* = 0.001, respectively). Concerning the physical practice variable, Student’s t-test showed higher values in autonomy in participants practicing PA compared with those not practicing PA (*p* = 0.001). However, no significant differences were detected between the two groups in the competence and relatedness variables (*p* = 0.316 and *p* = 0.299, respectively).

Concerning the nature of PA practiced during the lockdown period of the COVID-19 pandemic (Table 3), a one-way ANOVA showed significant effects of physical activity on autonomy, competence, and relatedness (F = 11.706, *p* = 0.001; F = 7.572, *p* = 0.001; F = 3.422, *p* < 0.005, respectively). The Bonferroni post hoc analysis showed higher autonomy values recorded in participants practicing PA (walking, aerobic exercise, muscular training, specialty training, and other activities) compared with those not practicing physical activity (*p* = 0.001 for all comparisons). Concerning the competence variable, higher values were recorded by participants practicing PA: walking (*p* = 0.001), aerobic exercise (*p* = 0.01), specialty training (0.001), and other activities (*p* = 0.001). However, regarding the relatedness variable, differences were recorded only between the participants practicing walking and those not practicing PA (*p* = 0.001). 

### 3.3. Effect of the Variable of Sleep Hours

The one-way ANOVA showed the significant effects of the hours of sleep variable on autonomy, competence, and relatedness (F = 13.965, *p* = 0.001; F = 11.652, *p* = 0.001; F = 11.974, *p* = 0.002, respectively). The Bonferroni post hoc test showed lower autonomy, competence, and relatedness values recorded in the group with less than 5 h of sleep per day compared with the other groups with sleep hours between 6 and 9 (*p* = 0.001) and with the one with sleep hours more than 10 (*p* = 0.001).

### 3.4. Effect of the Variable of University

Concerning autonomy, the one-way ANOVA showed a significant effect of the university variable (F = 3.172, *p* = 0.043). The Bonferroni post hoc test showed higher values for TU students compared with KSU students (*p* = 0.038). However, no significant differences were recorded between TU and KFU students (*p* = 0.391) and KSU and KFU students (*p* = 0.330). Concerning competence, the one-way ANOVA showed a significant effect of the university variable (F = 21.268, *p* = 0.001). 

The Bonferroni post hoc test showed higher competence perceptions in TU students compared with KFU students (*p* = 0.017) and KSU students (*p* = 0.001). Likewise, KFU students showed higher competence perceptions compared with KSU students (*p* = 0.004). Regarding relatedness, the one-way ANOVA showed a significant effect of the university variable (F = 25.058, *p* = 0.001). The Bonferroni post hoc test showed higher values for TU students compared with KFU students (*p* = 0.002) and KSU students (*p* = 0.001). Likewise, KFU students showed higher relatedness perceptions compared with KSU students (*p* = 0.005).

### 3.5. Effect of the Variable of Experience in Online Learning

Concerning the experience of online learning, the Student’s t-test comparison showed significantly higher autonomy values in the group claiming to have previous experience with online learning (*p* = 0.001). Similar results were found regarding the competence and relatedness variables with higher values in those groups with previous online learning experience (*p* = 0.001 and *p* = 0.014, respectively) (Table 4).

### 3.6. Effect of the Variable of GPA

Concerning the effect of the GPA variable, the results showed higher values of autonomy (*p* = 0.001), competence (*p* = 0.001), and relatedness (*p* = 0.002) in the group with GPAs above 3 to 5 compared with the group with GPAs ranging from 1 to 3. 

### 3.7. Correlation between Autonomy, Competence, Relatedness, and the Studied Variables 

We also examined the relationship between the three PNS variables (i.e., autonomy, competence, and relatedness) and all the independent variables (i.e., gender, experience in online learning, GPA, hours of sleep, PA practice, type of PA, and frequency of PA practice). The results showed high correlations between autonomy, competence, and relatedness. Likewise, there were high correlations between all three PNS variables and gender, experience in online learning, daily hours of sleep, and type of PA practiced (Table 5).

## 4. Discussion

This investigation aimed to (i) compare students’ PNS variables (autonomy, competence, and relatedness) in online PE lessons undertaken during the COVID-19 pandemic lockdown, and (ii) evaluate the relationship between participants’ autonomy, competence, and relatedness, as well as with the variables of gender, experience in online learning, university, GPA, hours of sleep, PA practice, and type of PA. The results showed a higher level of autonomy, competence, and relatedness in female participants compared with male participants; in TU students compared with KFU and KSU students; and in the group with previous online learning experience compared with the group with no online learning experience. Higher autonomy and competence levels were recorded in the group practicing physical activity (i.e., walking, aerobic exercise, muscular training, and specialty training) compared with the no-activity group, as well as in the group that slept more than six hours per day. High correlations were also observed between autonomy, competence, and relatedness, as well as the variables of gender, experience with online learning, hours of sleep, and type of PA, but not with GPA or frequency of PA.

The higher autonomy, competence, and relatedness perception levels recorded in females compared with males and in TU students compared with KFU and KSU students may be explained by the high level of motivation in females. Previous studies suggested that females are more autonomously regulated and males are more externally regulated regarding exercise behavior [31]. However, Teixeira et al. [32] showed that females have a general tendency toward more controlled regulations than men. In a meta-analysis, Guérin et al. [33] found negligible gender differences in motivational regulations. 

PE programs for females at Saudi universities were recently approved according to the Kingdom’s strategies to achieve its 2030 vision [34]. This might have affected the intrinsic motivation of Saudi women. Intrinsic motivation is considered the most self-determined, or autonomous, form of motivation. Ryan and Deci [20] argued that having satisfactory levels for all three PNS variables is required to enhance intrinsic motivation and well-being in general. When intrinsically motivated, a person performs a behavior volitionally because it feels inherently interesting or enjoyable [35]. As the TU participants were all females, the high level of PNS in comparison to KFU and KSU students, where participants were both males and females, makes sense.

The higher autonomy, competence, and relatedness levels recorded in the group with previous experience in online learning demonstrated the need to have ICT skills. Indeed, students in Saudi Arabian universities received online learning in some optional subjects (i.e., university requirements) via the Blackboard platform. Others (especially females) may receive more online subjects via live video transmission. Thus, they may be more familiarized with the non-presence of a teacher in the classroom, which represents an additional factor that affected the satisfaction of their psychological needs in online learning during the pandemic lockdown. Even though online learning is principally theoretical and does not let students practice and learn effectively [7], it provides an environment that allows students to be anywhere (independent) when learning and interacting with instructors and other students [36], which can affect their perception of autonomy, competence, and relatedness.

All of the PNS variables were shown to be affected by sleep habits and the type of PA practiced. Indeed the present findings showed higher values of autonomy, competence, and relatedness in the group of students that slept for more than six hours per day and in the group practicing PA (i.e., walking, aerobic exercise, muscular training, and other) in comparison to the group that had less than five hours of sleep and the group with no PA practice, respectively. The high correlations between autonomy, competence, and relatedness indicated the high relationships between these variables and their interdependence within PNS, which verified recent findings that reported high correlations between autonomy, competence, and relatedness in PE students [17,37] and adult active members [35]. Thus, as reported in other research [37,38], we can conclude that changes in one of the psychological needs induce changes in the others in PE students.

Positive correlations were detected between the three variables of psychological needs satisfaction and the variables of gender, experience in online learning, hours of sleep, and the type of PA undertaken during the COVID-19 pandemic lockdown; however, no correlation was detected with GPA and the frequency of PA. This result verified the findings of recent research [37], which showed that competence is strongly related, autonomy is moderately to strongly related, and relatedness is moderately related to enjoyment in PE. In a previous study [35], the presence of a high correlation was observed between the three psychological needs (autonomy, competence, and relatedness) and exercise intensity (moderate to strenuous) [18,39]. Therefore, we can conclude that the psychological need satisfactions are more related to the type of activity and to the intensity (moderate to strenuous) in which participants find their enjoyment, rather than its weekly frequency. Moreover, it was demonstrated that (specifically related to the SDT model and PA contexts) higher degrees of autonomy, competence, and relatedness are associated with increased exercise through self-determined motivation [40,41]. In the same manner, Edmunds et al. [42] found that the relationship between the need for competence and strenuous exercise was partially mediated by self-determined motivation.

## 5. Strengths and Limitations

The strength of this study lay mainly in the representative sample used and in the use of an Arabic version of the PE-ARCS of Sulz et al. [28]. Likewise, it assessed the relationship between PNS and factors that could affect PE students during the online learning performed during the COVID-19 pandemic lockdown. Nonetheless, some limitations merit discussion. The study design was cross-sectional and therefore not suitable for conclusions concerning causal links between the study variables. This study did not assess the relationship between psychological needs and PE students’ enjoyment during online lessons [37,43] or the effect of synchronous versus asynchronous online learning environments [44], which may represent possible future orientations for research. Moreover, it would be interesting to compare PE student profiles of PNS during and post-pandemic lockdown to better understand the influence of the restriction measures and the online learning. This could provide important information to improve both the online and the face-to-face learning strategies. 

## 6. Conclusions

Even with the aforementioned limitations, this study showed that a variety of variables could affect students’ autonomy, competence, and relatedness perceptions during online PE lessons. PNS is related to gender, experience in online learning, GPA, sleep habits, and type of PA. Walking, aerobic exercise, muscular training, and training in a specialty affected both autonomy and competence perceptions. However, relatedness was mainly affected by walking. Therefore, it is necessary to support ICT knowledge of students with low GPA and to encourage them to adopt balanced sleep and physical activity habits to increase their perceptions of autonomy, competence, and relatedness in online PE lessons. These findings may be useful for further theoretical development and PE teachers in structuring online learning content.

## Figures and Tables

**Table 1 ijerph-19-15288-t001:** Sociodemographic characteristics of the respondents (*n* = 321).

Variables	Values *
Gender	
Male	160 (49.8%)
Female	161 (50.2%)
Age	22.1 ± 3.35 years
Body mass index (BMI)	
Male	24.48 ± 0.28 kg/m^2^
Female	20.19 ± 0.22 kg/m^2^
Underweight (<18.5 kg/m^2^)	49 (15.3%)
Normal weight (18.5–24.9 kg/m^2^)	147 (45.8%)
Overweight (25–29.9 kg/m^2^)	38 (11.8%)
Obese (>30 kg/m^2^)	7 (2.2%)
Universities	
King Faisal University (KFU)	83 (25.9%)
King Saud University (KSU)	84 (26.2%)
Taif University (TU)	154 (48.9%)
Previous experience in online learning	
Yes	95 (29.6%)
No	226 (70.4%)
Grade point average (GPA)	
3 < GPA ≤ 5	213 (66.4%)
1 < GPA ≤ 3	108 (33.6%)

* Values are given as *n* (%) unless otherwise stated.

**Table 2 ijerph-19-15288-t002:** Effect of the variables of gender and physical practice on autonomy, competence, and relatedness.

			N	Mean ± SD	Student’s t	Sig.	95% CI
							L–U
Autonomy	Gender	Male	160	14.40 ± 3.045	2.096	0.037 *	–1.26 –0.040
		Female	161	15.05 ± 2.482			
	PA	Yes	285	14.94 ± 2.666	4.029	0.001 ***	0.995–2.893
		No	36	13.00 ± 3.180			
Competence	Gender	Male	160	15.21 ± 3.591	5.791	0.001 ***	–2.686–1.324
		Female	161	17.22 ± 2.522			
	PA	Yes	285	16.32 ± 3.077	1.171	0.243	–0.446–1.759
		No	36	15.67 ± 3.832			
Relatedness	Gender	Male	160	15.06 ± 2.965	6.580	0.001 ***	–2.453–1.323
		Female	161	16.95 ± 2.106			
	PA	Yes	285	16.07 ± 2.707	1.122	0.299	–0.409–1.494
		No	36	15.53 ± 2.942			

Significantly different at * *p* < 0.05 or *** *p* < 0.001.

**Table 3 ijerph-19-15288-t003:** Effect of the variable of physical activity practiced on autonomy, competence, and relatedness.

	Groups	N	Mean ± SD	95% CI	Mean Squared	F	Sig.
				L–U			
Autonomy	(I)	121	15.06 ± 2.296 ***	14.64–15.47	78.157	11.706	0.001
	(II)	31	15.35 ± 1.992 ***	14.62–16.09			
	(III)	76	14.91 ± 3.188 ***	14.18–15.64			
	(IV)	43	15.26 ± 2.564 ***	14.47–16.05			
	(V)	14	15.71 ± 1.637 ***	14.77–16.66			
	(VI)	36	11.67 ± 2.818	10.71–12.62			
Competence	(I)	121	16.60 ± 2.639 ***	16.10–17.09	67.130	7.572	0.001
	(II)	31	17.03 ± 3.180 ***	16.06–18.00			
	(III)	76	15.59 ± 2.617 *	14.87–16.32			
	(IV)	43	17.23 ± 1.834 ***	16.43–18.04			
	(V)	14	18.14 ± 4.067 ***	17.08–19.20			
	(VI)	36	14.03 ± 4.067	12.65–15.40			
Relatedness	(I)	121	16.37 ± 3.021 ***	15.95–16.80	24.657	3.422	0.005
	(II)	31	16.29 ± 2.550	15.42–17.16			
	(III)	76	15.91 ± 2.023	15.22–16.60			
	(IV)	43	16.14 ± 3.449	15.35–16.92			
	(V)	14	16.64 ± 2.023	15.47–17.81			
	(VI)	36	14.36 ± 3.449	13.19–15.53			

(I) Walking, (II) aerobic exercise, (III) muscular training, (IV) training in a specialty, (V) other activities, and (VI) no PA. Significantly different to VI at * *p* < 0.05 or *** *p* < 0.001.

**Table 4 ijerph-19-15288-t004:** Effect of previous experience in online learning on autonomy, competence, and relatedness.

	Experience with Online Learning	N	Mean ± SD	Student’s t	Sig.	95% CI (L–U)
Autonomy	Yes	95	15.92 ± 2.901 ***	5.145	0.001	1.044–2.336
	No	226	14.23 ± 2.592			
Competence	Yes	95	17.14 ± 3.254 ***	3.289	0.001	0.498–1.980
	No	226	15.90 ± 3.004			
Relatedness	Yes	95	16.59 ± 2.991 *	1.122	0.014	0.171–1.477
	No	226	15.77 ± 2.588			

Significantly different from the opposite response at * *p* < 0.05 or *** *p* < 0.001.

**Table 5 ijerph-19-15288-t005:** Relationship between autonomy, competence, relatedness, and all the independent variables.

		Autonomy	Competence	Relatedness	Gender	Experience with Online Learning	GPA	Sleep Hours	PA Practice	Type of PA	Frequency of PA Practice
Autonomy	Pear. correl.	1	0.699 **	0.459 **	0.117 *	–0.263 **	0.033	0.258 **	–0.220 **	–0.252 **	0.077
	Sig. (2-tailed)		0.000	0.000	0.037	0.000	0.560	0.000	0.000	0.000	0.171
	N	321	321	321	321	321	321	321	321	321	321
Competence	Pear. correl.	0.699 **	1	0.565 **	0.316 **	–0.267 **	0.017	0.227 **	–0.068	–0.149 **	0.075
	Sig. (2-tailed)	0.000		0.000	0.000	0.000	0.755	0.000	0.224	0.008	0.181
	N	321	321	321	321	321	321	321	321	321	321
Relatedness	Pear. correl.	0.459 **	0.565 **	1	0.346 **	–0.174 **	0.005	0.248 **	–0.063	–0.170 **	0.015
	Sig. (2-tailed)	0.000	0.000		0.000	0.002	0.930	0.000	0.263	0.002	0.795
	N	321	321	321	321	321	321	321	321	321	321
Gender (male = 1, female = 2)	Pear. correl.	0.117 *	0.316 **	0.346 **	1	–0.121 *	–0.170 **	0.088	–0.120 *	–0.147 **	0.051
	Sig. (2-tailed)	0.037	0.000	0.000		0.030	0.002	0.117	0.032	0.008	0.366
	N	321	321	321	321	321	321	321	321	321	321
Experience with online learning (yes = 1, no = 2)	Pear. correl.	–0.263 **	–0.267 **	–0.174 **	–0.121 *	1	0.582 **	–0.006	0.039	0.026	–0.014
	Sig. (2-tailed)	0.000	0.000	0.002	0.030		0.000	0.919	0.481	0.643	0.804
	N	321	321	321	321	321	321	321	321	321	321
GPA (3 < GPA ≤ 5 = 1, 1 < GPA ≤ 3 = 2)	Pear. correl.	0.033	0.017	0.005	–0.170 **	0.582 **	1	0.039	–0.023	–0.101	0.059
	Sig. (2-tailed)	0.560	0.755	0.930	0.002	0.000		0.490	0.678	0.072	0.295
	N	321	321	321	321	321	321	321	321	321	321
Hours of sleep (<5 = 1, 6–9 = 2, >10 = 3)	Pear. correl.	0.258 **	0.227 **	0.248 **	0.088	–0.006	0.039	1	–0.079	–0.032	–0.050
	Sig. (2-tailed)	0.000	0.000	0.000	0.117	0.919	0.490		0.160	0.571	0.374
	N	321	321	321	321	321	321	321	321	321	321
PA practice (yes = 1, no = 2)	Pear. correl.	–0.220 **	–0.068	–0.063	–0.120 *	0.039	–0.023	–0.079	1	0.513 **	–0.218 **
	Sig. (2-tailed)	0.000	0.224	0.263	0.032	0.481	0.678	0.160		0.000	0.000
	N	321	321	321	321	321	321	321	321	321	321
Type of PA (I, II, III, IV, V, VI)	Pear. correl.	–0.252 **	–0.149 **	–0.170 **	–0.147 **	0.026	–0.101	–0.032	0.513 **	1	–0.083
	Sig. (2-tailed)	0.000	0.008	0.002	0.008	0.643	0.072	0.571	0.000		0.139
	N	321	321	321	321	321	321	321	321	321	321
Frequency of PA practice	Pear. correl.	0.077	0.075	0.015	0.051	–0.014	0.059	–0.050	–0.218 **	–0.083	1
	Sig. (2-tailed)	0.171	0.181	0.795	0.366	0.804	0.295	0.374	0.000	0.139	
	N	321	321	321	321	321	321	321	321	321	321

(I) Walking, (II) aerobic exercise, (III) muscular training, (IV) training in a specialty, (V) other activities, and (VI) no PA. * Correlation is significant at *p* < 0.05.; ** Correlation is significant at *p* < 0.01.

## Data Availability

All datasets used and/or analyses during the current study are available from the corresponding author upon reasonable request.

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
