# Peer review of "Students’ Perceptions in Online Physical Education Learning: Comparison Study of Autonomy, Competence, and Relatedness in Saudi Students during the COVID-19 Lockdown"

_ijerph, 2022, doi:10.3390/ijerph192215288_

Round 1

Reviewer 1 Report

Thank you very much for the opportunity to review this study. According to my point of view, there are some issues that would improve the quality of this manuscript as the following:

Abstract:

-        A brief introduction is needed, and some recommendations for practice based on this study’s findings should be provided.

-       All acronyms have to be spelled out the first time they are used (e.g., PA, PE, KSU, TU, KFU) throughout this paper, including the “abstract” section.

Methods:

-        How many potential respondents declined to answer the survey? Please provide the response rate of the current study. Did they occur in specific questions? If cases with missing responses were to be completely excluded from the analytic sample, would there have been sampling bias altogether?

-        Instruments: How to assess the sleeping and physical activity habits of the participants? More details about items and psychrometric properties are needed.  

Results:

-        The authors should move “Table 1” from the “methods” section to the “Results” section. It would be helpful for the readers if the authors could provide the participants' characteristics or demographic data to understand clearly about physical education students in this study.

-        The response rate of the final sample comprised needs to be mentioned in this survey.

-        All exact significance levels of the study’s findings should be presented instead of p < 0.05, p < 0.001, and p < 0.001. For example, “Concerning the variable gender, the statistical analysis showed higher values in female compared with male students on autonomy (p = 0.037), competence (p < 0.001), and relatedness (p < 0.001) (Lines 157-159). Also, a one-way ANOVA showed significant effects of physical activity on autonomy, competence, and relatedness (F = 11.706, p = 0.001; F = 7.572, p = 0.001; F = 3.422, p = 0.005, respectively). Please apply this to all findings.

-        Ethical Considerations in Lines 116-117 should be deleted because the authors already stated in the “Institutional Review Board Statement” section (Lines 328-329); however, the approval date needs to be mentioned.

Good luck!

Reviewer 2 Report

The study has been conducted very properly. It is regrettable, then, that the published paper is not likely to invite substantial scientific interest due to its unstimulating subject. I suggest the authors to repeat the study with a new sample from the same population and compare the current and new data for effect sizes, thus possibly revealing important novel findings. Furthermore, the new design might shed light on questions such as whether and to what extent the pandemic influenced the level of perceived self-determination needs among university students of physical education.

Reviewer 3 Report

The study examines a current topic: This investigation aimed to compare students’ PNS variables (autonomy, competence, and relatedness) in online physical education lessons undertaken during the COVID-19 pandemic lockdown, and evaluate the relationship between participants’ autonomy, competence, and relatedness, as well as with the variables gender, experience in online learning, university, GPA, hours of sleep, PA practice, and type of PA.

The literature background raises some problems related to online education and highlights some advantages of traditional education. This approach is appropriate for posing a problem. The authors used literature sources from the last three decades. This chapter can still be developed.

The chosen method is quantitative, which is suitable for understanding students' behavior.

The design of the database was correct. We do not receive information about the criteria on which 249 participants were excluded from the study. I propose a description of this process.

The derivation of the results is logical. The conclusions are correct.

In the chapter 'Discussion', the authors compare their own results with the results of previous research, as required by scientific analysis.

The tables are well structured.

The study also has findings beyond the specific investigation, which support the results of many previous studies: „. It is necessary to encourage students to improve their ICT experience and to adopt balanced sleep and physical activity habits to increase their perceptions of autonomy, competence, and relatedness.”

I also found some formal errors, e.g. in the chapter title of point 5.

In summary, I accept the manuscript after minor revision.

Round 2

Reviewer 2 Report

Provided that the above points of criticism are enumerated under the limitations of the study, I approve of the publication of the manuscript.